# Comparative Transcriptome and Metabolome Profiling Reveal Mechanisms of Red Leaf Color Fading in *Populus* × *euramericana* cv. ‘Zhonghuahongye’

**DOI:** 10.3390/plants12193511

**Published:** 2023-10-09

**Authors:** Shaowei Zhang, Xinran Yu, Mengjiao Chen, Cuifang Chang, Jingle Zhu, Han Zhao

**Affiliations:** 1Research Institute of Non-Timber Forestry, Chinese Academy of Forestry, 3 Weiwu Road, Zhengzhou 450003, China; xinranyu54@163.com; 2College of Rural Revitalization, The Open University of Henan, 36 Longzi Lake North Road, Zhengzhou 450046, China; 3Research Institute of Tropical Forestry, Chinese Academy of Forestry, 682 Guangshan 1st Road, Guangzhou 510520, China; cmjmengjiao1997@163.com; 4The College of Landscape Architecture and the Arts, Henan Agricultural University, 63 Agricultural Road, Zhengzhou 450002, China; changcuifang0709@163.com

**Keywords:** anthocyanin, the color fading of red leaves, metabolome, transcriptome, molecular breeding

## Abstract

Anthocyanins are among the flavonoids that serve as the principal pigments affecting the color of plants. During leaf growth, the leaf color of ‘Zhonghuahongye’ gradually changes from copper-brown to yellow-green. At present, the mechanism of color change at different stages has not yet been discovered. To find this, we compared the color phenotype, metabolome, and transcriptome of the three leaf stages. The results showed that the anthocyanin content of leaves decreased by 62.5% and the chlorophyll content increased by 204.35%, 69.23%, 155.56% and 60%, respectively. Differential metabolites and genes were enriched in the pathway related to the synthesis of ‘Zhonghuahongye’ flavonoids and anthocyanins and to the biosynthesis of secondary metabolites. Furthermore, 273 flavonoid metabolites were detected, with a total of eight classes. *DFR*, *FLS* and *ANS* downstream of anthocyanin synthesis may be the key structural genes in reducing anthocyanin synthesis and accumulation in the green leaf of ‘Zhonghuahongye’. The results of multi-omics analysis showed that the formation of color was primarily affected by anthocyanin regulation and its related synthesis-affected genes. This study preliminarily analyzed the green regression gene and metabolic changes in ‘Zhonghuahongye’ red leaves and constitutes a reference for the molecular breeding of ‘Zhonghuahongye’ red leaves.

## 1. Introduction

*Populus nigra*, an important *Populus* species, has excellent characteristics such as strong adaptability, short rotation period, being asexual, and demonstrating simple sexual reproduction. This is widely used in biofuels, papermaking, ecological restoration, shelterbelt afforestation, etc. [1]. As one of the most widely distributed deciduous broad-leaved trees in the world, *Populus nigra* is primarily distributed in Asia, Europe, North America, Middle Eastern countries and Mediterranean coastal areas [2]. ‘Zhonghuahongye’ (*populus* × *euramericana* cv.‘Zhonghuahongye’), formerly known as Chinese red poplar, is a kind of poplar with a dark red leaf color, obvious seasonal change, and excellent landscape effects. It is a good variety of colored leaf ornamental wood independently cultivated in China [3]. However, in the process of the large-scale reproduction of ‘Zhonghuahongye’, a small number of green leaves were found in the red leaf population. These leaves of ‘Zhonghuahongye’ were slightly red in the young period, and then were green as a whole. This phenomenon is widespread in many plants in the subtropics and tropics. In recent years, the research on the formation mechanism of the color fading of red leaf phenomena in the plant kingdom has gradually attracted attention [4]. As for the mechanism of leaf color change, many studies have indicated that the red color of leaves is related to the change in pigment content, anthocyanins, and candidate genes. The attenuation of ‘Zhonghuahongye’ may be due to the change in related pigment content during development.

Atavism, in which organisms that have been domesticated in some cases accidentally develop some genetic characteristics of their ancestors, is generally considered a “degenerate” phenomenon. Though uncommon, it is prevalent in some species in nature [5]. When seeds produced via sexual reproduction are raised, some plants exhibit varying degrees of atavism [6]. The report of Song et al. [7] provided a theoretical explanation for the phenomenon of “atavism” in Saururus chinensis, i.e., ontogenetic color change from green before flowering into white during flowering, and then reversion back to green during fruit development. ‘Zhonghuahongye’ was obtained from the selection of 2025 poplar (green leaves) of *Populus nigra* [8]. The color fading of its red leaves may be an atavistic phenomenon. Various atavistic phenomena in the plant kingdom indicate that some genes controlling original ancestor traits are not lost during phylogeny but are merely inactivated [9]. The color fading of red leaves may occur due to some environmental factors that activate genes that have been inactivated during phylogeny.

Previous studies on leaf color changes in ‘Zhonghuahongye’ have primarily focused on leaf pigment changes and metabolite changes [10]. Although metabonomic studies for flavonoids have been conducted using ‘Zhonghuahongye’ [11], there has been limited research focusing on flavonoid biosynthesis. Additionally, the molecular mechanism of flavonoid accumulation and biosynthesis in ‘Zhonghuahongye’ remains largely unexplored. In this study, the transcriptome and metabolome of ‘Zhonghuahongye’ were investigated. The results of high-performance liquid chromatography showed that anthocyanin content decreased significantly during the process of shifting from red to green leaves. Then, we constructed the metabolic regulatory network of anthocyanin synthesis by measuring the expression level of differential genes. We screened the key functional genes and transcription factors that led to the shift from red to green in leaves, which provided important insights for the further study of the anthocyanin biosynthesis regulatory network. Meanwhile, our investigation provided new ideas for the study of atavistic phenomena and provided a technical theoretical basis for modern poplar molecular breeding.

## 2. Results

### 2.1. Comparison of Color and Phenotype and Changes in Leaf Pigment Content

According to the leaf phenotype and microstructure analysis, the leaf shape of ‘Zhonghuahongye’ was oval, and the petiole, moving from leaf to branch, changed from cylindrical to lateral flat. As shown in Figure 1a, the leaves of ‘Zhonghuahongye’ are coppery-brown in the G1 period, greenish-brown in the G2 period, and fern-green in the G3 period.

According to the comprehensive analysis of leaf profile and microstructure, the color distribution of leaves is not uniform. The leaf petiole of ‘Zhonghuahongye’ is fuzzy, the leaves turn from red to green and the palisade tissue of mesophyll cells turns from red to green during leaf maturation. The mesophyll cells showed that the leaves of the ‘Zhonghuahongye’ showed a double-layer palisade tissue. The content of petiole anthocyanins increased after a little decrease. Anthocyanins increased first and then decreased. In the G1 stage, the content of anthocyanins was abundant. In the G2 stage, there was a small quantity of anthocyanins. In the G3 stage, anthocyanins almost disappeared. At the stage of young leaves, the anthocyanin content in each tissue was the highest. Conversely, with the growth of leaves, the anthocyanin content gradually decreased, while the petiole anthocyanin content was almost unchanged (Figure 1b).

As shown in Table 1, from G1 to G3, anthocyanin content decreased from 0.24 to 0.09 with a change range of 62.5% and the content of chlorophyll a increased from 0.23 to 0.70 with a change range of 204.35%. The content of chlorophyll b increased from 0.13 to 0.22 (+69.23%). The content of the total chlorophyll increased from 0.36 to 0.92 with a variation range of 155.56%. The content of carotenoid increased from 0.15 to 0.24 (+60%). The contents of carotenoids and anthocyanins in leaves are low, and the variation range is also small. According to the leaf pigment analysis, the primary substances that make the leaves of ‘Zhonghuahongye’ red are anthocyanins. The phenomenon whereby leaves turn from red to green occurs due to a sharp decrease in anthocyanin content and an increase in chlorophyll content.

### 2.2. Qualitative and Quantitative Analysis of Leaf Metabolites in Different Periods

In order to further analyze the reasons for the phenomenon whereby leaves of ‘Zhonghuahongye’ turn from red to green, cluster analysis was conducted on the total metabolites of leaves of ‘Zhonghuahongye’ at different periods. The results showed that there was an obvious separation trend between groups. The samples in the group had good reproducibility. The proposed model was stable and reliable, which made it suitable for this experiment. The metabolic spectrum clustering was performed based on leaf color during the different periods and produced clear results (Figure 2a). In this experiment, liquid chromatography–tandem mass spectrometry (LC-MS/MS) was used to complete the qualitative and quantitative determination of metabolites, and the PCA differences were preliminarily verified. In particular, the first, second and third principal components accounted for 29.8%, 25.89% and 12.71% of the total variables (Figure 2b). As shown in Figure 2c, a total of 273 metabolites were identified in the G1, G2 and G3 systems and primarily divided into eight categories. The flavonoids and flavonols were the most diverse, respectively (114 and 41), followed by flavonoids (34), flavanones (25), anthocyanins (21), polyphenols (18), isoflavones (15) and proanthocyanins (5). Among these, flavonoids are related to leaf color.

### 2.3. Analysis of Leaf Metabolites Difference in Different Periods

According to the results of orthogonal partial least-squares discriminant analysis (OPLS-DA) (Appendix A), combined with the screening criteria for differential metabolites adopted in this study, the samples were divided into G1 and G2 (G1 vs. G2), G1 and G3 (G1 vs. G3), and G2 and G3 (G2 vs. G3). The differential metabolites were compared. As shown in Figure 3a,b, there were 64 differential metabolites (26 up-regulated and 38 down-regulated), 80 differential metabolites (38 up-regulated and 42 down-regulated) and 43 differential metabolites (21 up-regulated and 22 down-regulated) in G1 vs. G2, G1 vs. G3 and G2 vs. G3 groups, respectively. The difference between metabolites in G1 and G2 periods is significantly greater than that in G2 and G3 periods. It is interesting that most of them are down-regulated.

Venn diagram (Figure 3c) analysis showed that G1 vs. G2, G1 vs. G3, and G2 vs. G3 had a total of nine significant differential metabolites, and that the number of characteristic differential metabolites in G2 vs. G3 was significantly higher than that in G1 vs. G2. As shown in the histogram of the multiple differences (Figure 3d), the metabolites that differ among the three groups include eight types of metabolites, namely, anthocyanins, flavonoids, flavonols, flavanones, flavonoids, proanthocyanins, isoflavones and polyphenols. Because of the affection of the leaves’ color, these flavonoids are considered as the “core metabolite group” of the leaves of ‘Zhonghuahongye’. As shown in Figure 3c,d, among the top 10 metabolites with the largest down-regulated multiples in G1 vs. G2 group, 9 were flavonoids and anthocyanins. Among the 64 differential metabolites, 5 polyphenols, 11 anthocyanins, 10 flavonols, 31 flavonoids, 8 flavonoids, 7 flavanones, 5 isoflavones and 1 proanthocyanins were included. In the G1 vs. G3 group, the top 10 metabolites with the largest down-regulation ratio were flavonoids and anthocyanins, and the 80 substances included 1 polyphenol, 8 anthocyanins, 7 flavonols, 27 flavonoids, 7 flavonoids, 8 flavanones, 6 isoflavones and 2 proanthocyanins. In G2 vs. G3 group, the top 10 metabolites with the largest down-regulation ratio were flavonoids and anthocyanins, and the 43 substances included 4 polyphenols, 4 anthocyanins, 5 flavonols, 13 flavonoids, 5 flavonoids, 8 flavanones, 3 isoflavones and 1 proanthocyanins. The down-regulated expression of anthocyanins and flavonoids primarily occurred during the G1-to-G2 transition period.

### 2.4. Functional Annotation and Enrichment Analysis of Differential Metabolite KEGG

Since flavonoids were the major differential metabolites, the detected metabolites were annotated using the KEGG database, and then the annotation results were classified according to the type of pathway in KEGG. According to the analysis (Appendix A), differential metabolites were primarily concentrated in four metabolic pathways: flavonoid and flavonol biosynthesis (ko00944), flavonoid biosynthesis (ko00941), isoflavone biosynthesis (ko00943) and anthocyanin biosynthesis (ko00942). At different stages of green leaf development, the differential metabolites were primarily related to the flavonoid biosynthesis pathway, and also to the isoflavone biosynthesis pathway. The total concentration of G1 vs. G3 in the anthocyanin biosynthesis pathway had more significant differential metabolites. Differential metabolites in G1 vs. G2, G1 vs. G3, and G2 vs. G3 were significantly enriched in the biosynthetic pathways of flavonoids and flavonols. The number of annotated differential metabolites in each KEGG pathway was shown in Table 2. These results indicate that the phenotypic changes are dominated by the biosynthetic pathways of flavonoids and flavonols at different developmental stages of the green leaves of ‘Zhonghuahongye’, and the anthocyanin biosynthesis pathway plays a dominant role in the phenotypic changes in G1 vs. G3.

KEGG enrichment analysis showed (Appendix A) that secondary differential metabolic pathways were significantly enriched in anthocyanin biosynthesis pathways. In the G1 vs. G2 group, eight anthocyanins were identified. These included peonidin *O*-hexoside, cyanidin 3-*O*-glucosyl-malonylglucoside, rosinidin *O*-hexoside, cyanidin 3-*O*-glucoside (kuromanin), delphinidin *O*-malonyl-malonylhexoside, malvidin 3-*O*-galactoside, cyanidin, and peonidin 3-*O*-glucoside chloride.

In the G1 vs. G3 group, 11 anthocyanins were identified (peonidin *O*-hexoside, cyanidin 3-*O*-glucosyl-malonylglucoside, rosinidin *O*-hexoside, cyanidin 3-*O*-glucoside (kuromanin), cyanidin *O*-syringic acid, malvidin 3-*O*-galactoside, cyanidin 3,5-*O*-diglucoside (cyanin), cyanidin, cyanidin 3-*O*-galactoside, peonidin 3-*O*-glucoside chloride, and apigeninidin chloride). In the G2 vs. G3 group, four anthocyanins were identified (cyanidin *O*-syringic acid, malvidin 3-*O*-glucoside (Oenin), cyanidin 3,5-*O*-diglucoside (cyanin), and peonidin 3-*O*-Glucoside chloride). To sum up, anthocyanins (cyanidin, peonidin, and malvidin) play important roles in the color fading of ‘Zhonghuahongye’.

In order to further understand the relationship between flavonoids and leaf color, transcriptome analysis was performed (Figure 4). By comparison, 4480 genes (1969 up-regulated, 2511 down-regulated), 6937 genes (3062 up-regulated and 3875 down-regulated), and 3871 genes (1772 up-regulated, 2099 down-regulated) (Figure 4a,b) were found in G1 vs. G2, G1 vs. G3, and G2 vs. G3, respectively. In order to study the expression patterns of genes in leaves at different periods, k-means cluster analysis was performed for the FPKM of genes (Figure 4c). The k-means cluster analysis showed that 4899 genes gradually declined and 4213 genes gradually increased. Different gene products in an organism play different biological functions through their interactions.

Among the genes, there were 780 significantly different genes between G1 vs. G2, G1 vs. G3, and G2 vs. G3 (Figure 4d,e). These differential genes are considered to be the “core transcriptome” of the leaves of ‘Zhonghuahongye’. The KEGG enrichment analysis of differentially expressed genes (Figure 4f) showed that differentially expressed genes were primarily enriched in two pathways, one for the biosynthesis of secondary metabolites (ko01110) and one with a metabolic function (ko01100). Comparing G1 vs. G2, we found some significant differences in the enrichment pathways, including phenylpropanoid biosynthesis (ko00940) and isoflavonoid biosynthesis (ko00941). In G1 vs. G3, we found some significant differences in the enrichment pathways including plant hormone signal transduction (ko04075), ribosome (ko03010), isoflavonoid biosynthesis (ko00943), anthocyanin biosynthesis (ko00942) and phenylpropanoid biosynthesis (ko00940). In G2 vs. G3, we found some significant differences in the enrichment pathways including anthocyanin biosynthesis (ko00942), pentose and glucuronate interconversions (ko00040), and flavonoid biosynthesis (ko00941).

### 2.5. Metabolome and Transcriptome Association Analysis

#### 2.5.1. Model-Based Association Analysis

According to metabolome PCA (Appendix A), the first and second principal components accounted for 40.94% and 17.19% of the total variables, respectively. According to transcriptome PCA (Appendix A), however, the first and second principal components accounted for 29.66% and 20.83% of the total variables, respectively. The difference between the three biological replicates was small. The quality control samples were gathered, and the data detection was stable. The metabolites and genes between the groups were separated, indicating that there were obvious differences between the sample groups at different developmental stages. Additionally, the sample groups were aggregated, indicating that the stability in the group was robust.

#### 2.5.2. Association Analysis Based on KEGG

Enrichment analysis of differential metabolites and differential genes showed that differential metabolites and differential genes were primarily enriched via six pathways (Table 3), including via the biosynthesis of secondary metabolites (ko01110), the metabolic pathway (ko01100), isoflavone biosynthesis (ko00943), anthocyanin biosynthesis (ko00942), flavone and flavonol biosynthesis (ko00944). Among them, the metabolic pathway with the most significant enrichment of differential metabolites in each group is the flavone and flavonol biosynthesis pathway, and this metabolic pathway is the most significant enrichment pathway for differential genes (Appendix A).

#### 2.5.3. Correlation Analysis Based on Correlation

As shown in Figure 5, operating according to the difference multiples of genes and metabolites with Pearson correlations coefficient greater than 0.8 in each difference group, the genes and metabolites were divided into 1–9 quadrants from left to right and from top to bottom with black dashed lines. In the quadrant 5, genes and metabolites were not differentially expressed. In quadrant 3 and 7, the differential expression patterns of genes and metabolites were consistent, and the regulation of genes and metabolites was positively correlated, and the change in metabolites might be positively regulated by genes. In quadrant 1, 9, the differential expression pattern of genes and metabolites was opposite, and the regulation of genes and metabolites was negatively correlated, and the change in metabolite expression might be negatively regulated by genes. In quadrants 2, 4, 6, and 8, metabolite expression remained unchanged, and genes were regulated up and down, or gene expression remains unchanged, and metabolites are regulated up and down. In order to find transcription factors affecting anthocyanin synthesis in the leaves of ‘Zhonghuahongye’, structural genes consistent with metabolite expression patterns should be focused on in quadrants 3 and 7, and related transcription factors or metabolites should be screened in quadrant 2, 4, 6, and 8.

#### 2.5.4. Influence on Differential Metabolites and Differential Genes Regulating Anthocyanin Synthesis of ‘Zhonghuahongye’

According to the correlation network diagram (Appendix A), in G1 vs. G2, naringenin 7-*O*-glucoside (prunin) (pme0371) was negatively correlated with aromadedrin (Dihydrokaempferol) (pme2963), cyanidin (pme3609) and three genes (Appendix A). Naringenin 7-*O*-glucoside (prunin) (pme0371) was positively correlated with bromo-adjacent homology domain-containing protein1 (BAHD1) (POPTR_010G056300v4). Dihydromyricetin (pme2898) was negatively correlated with two genes. POPTR_019G001300 was positively correlated with other genes and metabolites. In G1 vs. G3, there was a complex regulatory relationship between naringenin 7-*O*-glucoside (prunin) (pme0371), cyanidin (pme3609), dihydromyricetin (pme2898), butein (pme3439) and 22 genes. In G2 vs. G3, hesperetin (pme2319) was positively correlated with four genes. POPTR_009G128800 was negatively correlated with naringenin (pme0376), kaempferol (pme0200), 4,2′,4′,6′-tetrahydroxychalcone (pmf0057), garbanzol (pmf0108) and 4′,5,7-trihydroxyflavanone (pmf0058).

In Figure 6 (correlation cluster heat map), we conducted a comprehensive analysis of the up-regulation and down-regulation of differential metabolites related to G1 vs. G2, G1 vs. G3, and G2 vs. G3 color formation pathways. The results showed that the correlation coefficient between samples of the same species at different periods was high, while the correlation between different species was weak. This showed that the metabolites and genes in the leaves of different species were different. As can be seen from the correlation analysis chart, the correlation coefficient within the same group was close to 1, indicating that the sample repeatability was good. Meanwhile, the obtained differential metabolites and differential genes were relatively reliable.

#### 2.5.5. Screening of Related Transcription Factors and Related Metabolites Affecting Anthocyanin Synthesis in Leaves of ‘Zhonghuahongye’

Typical correlation analysis was carried out on differential genes and differential metabolites, as shown in Figure 7. Four regions were distinguished by crosses in the figure. In the same region, the farther away from the origin a point was, the higher the correlation would be. In the lower left region, the naringenin 7-*O*-glucoside (prunin)(pme0371) had a high association with the gene.

In G1 vs. G2, G1 vs. G3 and G2 vs. G3 combinations, 434 transcription factors were differentially expressed simultaneously, including 13 *B3*, 32 basic helix-loop-helix (*bHLH*), 18 basic leucine zipper (*bZIP*), 31 V-myb avian myeloblastosis viral oncogene homolog *(MYB*), 25 *NAC*, and 20 *WRKY* transcription factors. These 390 transcription factors are considered to be important factors in regulating anthocyanin synthesis at different developmental stages of the green leaves of ‘Zhonghuahongye’. Some of the genes are listed in Table 4. In most plants, the biosynthetic pathways of flavonoids and anthocyanins are primarily affected by the interaction of *MYB*, *bHLH*, *WD40*, *NAC*, and *bZIP5* transcription factors [12,13,14]. As mentioned earlier, 31 *MYB* transcription factors were identified in the genome of ‘Zhonghuahongye’. Combining sequence homology information with the sequence homology of *MYB* reported in *Arabidopsis thaliana*, *Populus trichocarpa*, *Vitis vinifera*, and *Zea mays*, a phylogenetic tree of *MYB* in ‘Zhonghuahongye’ that might regulate anthocyanin synthesis was constructed. (Appendix A). The results showed that the candidate *MYB* in this report is homologous to the *MYB* gene known to regulate flavonoid and anthocyanin synthesis, which are functionally similar, and be involved in the regulation of anthocyanin synthesis in ‘Zhonghuahongye’.

#### 2.5.6. Key Genes Were Verified Using qRT-PCR

In order to verify the reliability of the results of RNA-Seq, 10 genes were selected for qRT-PCR verification. As shown in Appendix A, the transcriptome data of the leaves of ‘Zhonghuahongye’ at various developmental stages were consistent with the results of qRT-PCR.

#### 2.5.7. Study on the Synthesis and Regulation of Anthocyanins in Leaves of ‘Zhonghuahongye’

Plant leaves have a wide variety of colors. These colors primarily occur due to four different classes of plant metabolites: chlorophyll, carotenoids, betaine, and flavonoids. According to the above determination, anthocyanins were the principal pigments involved in the changes in pigment content in the leaves of ‘Zhonghuahongye’ at different periods. Anthocyanins belong to flavonoids, which have the effect of making plants red. They are not only diverse in structure, but also appear in large quantities. The plant primarily has six kinds of non-ligands (aglycone), including pelargonidin, cyanidin, delphinidin, petunidin and malvidin. The synthesis and accumulation of anthocyanins determine the contents of anthocyanins in leaves. The primary pathways related to anthocyanin synthesis are flavonoid biosynthesis (ko00941) and anthocyanin biosynthesis (ko00942). Operating according to the results of differential metabolic enrichment, we focused on six pathways: biosynthesis of secondary metabolites (ko01110), metabolic pathway (ko01100), isoflavonoid biosynthesis (ko00943), anthocyanin biosynthesis (ko00942), flavonoid biosynthesis (ko00941), flavone biosynthesis, and flavonol biosynthesis (ko00944). Combined with the screening of differential metabolites related to the regulation of anthocyanin synthesis, the anthocyanin bioanabolic diagram was drawn according to the combined analysis results (Figure 8).

## 3. Discussion

### 3.1. Primary Study on the Main Substances of Leaf Color and the Cause of Color Fading of Red Leaves of ‘Zhonghuahongye’

Leaves, comprising the primary plant organ, display a given color as the result of a variety of factors, in which different factors jointly affect the accumulation of pigments to determine the leaves’ color. The color of plants with colored leaves can be explained by changes in the types and proportions of pigments in the leaves. The pigments in the leaves of higher plants can be classified into the following three categories: one is flavonoids, primarily anthocyanins; the second is carotenoids, which primarily includes lutein and carotenoid pigments; and the third category is chlorophyll, which primarily includes chlorophyll a and b.

Anthocyanins, a type of flavonoid, play important roles in the plant world due to their health benefits and environmental stress tolerance [15]. Additionally, they have been intensively studied for more than a century [16]. Anthocyanins are the principal pigments that affect the color and are the direct cause of the red color of leaves. Wang Jing [17] has reported that anthocyanins contribute directly to the color of leaves and that other non-anthocyanin substances assist in the color. In the present study, the change trends of chlorophyll and anthocyanins were mostly the opposite [18]. In our study, the leaves of ‘Zhonghuahongye’ were copper-brown in the early stages, with the highest content of anthocyanins in each tissue. With leaf growth of leaves, the color of leaves gradually turned to fern green, the content of anthocyanins gradually decreased, the content of chlorophyll increased significantly, and the anthocyanins of petioles remained almost unchanged. Therefore, we speculate that anthocyanins are the primary substances that make the leaves of the early stage take a copper-brown color and that chlorophyll is the primary substance that makes the leaves of the late-stage fern green. The extent of color fading of red leaves was directly determined by the decrease in anthocyanin content and the increase in chlorophyll content.

There are many kinds of anthocyanins in nature, including pelargonidin, cyanidin, delphinidin, peonidin, petunidin and malvidin. Different anthocyanins make plants take on different colors. For example, the primary anthocyanins that turn crape myrtle flowers red are mallurin and petunidin. The principal anthocyanins that make strawberry flowers red are cyanidin-3-*O*-glucoside and geranium-3-*O*-glucoside. Cyanidin-3-(6″-acetylgalactoside) and cyanidin-3-arabinoside in the leaves of Loropetalum chinense are the decisive factors in red color [19]. In the study of cherries, it was found that there was a significant correlation between the increase in the content of malvidin 3-*O*-glucoside (violet color) and pelargonidin 3-*O*-glucoside (orange red). At the same time, there was a change in leaf color from green to purple-red [20]. In this study, the contents of cyanidin 3-*O*-glucoside, peonidin *O*-hexoside and malvidin 3-*O*-galactoside in the growing leaves significantly decreased. So, we speculate that these three types of anthocyanins may be the cause of the disappearance of Zhonghuahongye’s red leaves.

The changes in anthocyanin content are related to the environment and growth stage. Previous studies have found that anthocyanins usually accumulate in the nutritional and reproductive parts of plants when exposed to adverse environmental factors during the young leaf stage. The leaves synthesize a large amount of anthocyanins in order to protect themselves from excessive ultraviolet radiation and reduce UV damage. At the same time, at the early stage of leaf growth, the temperature is low, the leaves are tender, and the ability to synthesize chlorophyll is weak. Anthocyanins often dominate in various pigments, and so the leaf color often appears red. As the leaves mature and leaf sizes increase, plants synthesize a large amount of chlorophyll to meet the needs of photosynthesis for plant growth, reducing the accumulation of anthocyanins [21]. Some scholars have proposed that anthocyanins can absorb green and ultraviolet light, reflect red and blue light, and play a role in filtering light and affecting photosynthetic capacity. Therefore, in the later stage of plant growth, the content of anthocyanins will decrease. When the temperature difference between day and night is large, the accumulation of sugar in the leaves increases, further promoting the synthesis of anthocyanins [22]. Our results indicate that, from G1 to G3, the anthocyanin content decreases and the chlorophyll content increases in the leaves of ‘Zhonghuahongye’. Therefore, we speculate that the phenomenon of chlorosis in the leaves of ‘Zhonghuahongye’ occurs due to a decrease in anthocyanin content and an increase in chlorophyll content as the leaves mature to meet the needs of plant growth.

Studies have shown that the decrease in anthocyanin content may be due to the degradation of anthocyanins and the termination of anthocyanin synthesis [23]. Meanwhile, it may also be accompanied by leaf maturation, leaf expansion, and the dilution of anthocyanins in tissues. When the climate is humid and the water content in trees increases, the concentration of anthocyanin decreases [24]. In our study, during the G1 period, the content of anthocyanins in the mesophyll is abundant; conversely, during the G2 and G3 periods, there are small amounts of anthocyanins. Therefore, we infer that as the leaf maturation of ‘Zhonghuahongye’ and the leaf tissue continue to expand, the anthocyanin concentration is diluted. At the same time, under the regulation of internal genes, the synthesis of anthocyanins is terminated and internal anthocyanins are gradually degraded.

### 3.2. Changes in Metabolome and Transcriptome in Leaves of ‘Zhonghuahongye’ in Different Periods

Transcriptome and metabolome data in different periods play important roles in studying transcriptome and metabolome changes during species development, helping to study species’ high-content metabolites and high-expression genes, analyzing the synthesis mechanisms of key metabolites, and identifying multiple genes involved in the regulation of important metabolite synthesis [25]. In this study, in order to further determine the reason for the decrease in anthocyanin content in the leaves of ‘Zhonghuahongye’ with growth, we measured the metabolome and transcriptome of the three stages.

Through transcriptome sequencing and metabolome detection of Zhonghuahongye’s leaves in three periods, this study obtained detailed metabolic and expression profile data of Zhonghuahongye’s leaves in different periods. Through in-depth analysis, we obtained the “core metabolome” and “core transcriptome” of Zhonghuahongye’s leaves. This not only furnished valuable data resources for this study, but also provided valuable data resources for research into other poplar species. The differential analysis of metabolome and transcriptome in the three periods showed that the differential metabolites were primarily concentrated in four metabolic pathways: flavone and flavonol biosynthesis (ko00944), flavonoid biosynthesis (ko00941), isoflavonoid biosynthesis (ko00943) and anthocyanin biosynthesis (ko00942). The differentially expressed genes are primarily enriched in the biosynthesis of secondary metabolites (ko01110) and metabolic pathways (ko01100), providing an important theoretical basis for studying the regulation of key traits and the synthesis of key metabolites.

The color of plant tissues and organs is primarily related to the conversion of flavonoids. The four major types of pigments that determine the color of plant tissues and organs are all secondary metabolites, and their synthesis involves multiple metabolic steps, making their mechanisms of action very complex. Previous studies have proposed that the conversion of flavonoids is also related to the color of leaves [26]. Cheng et al.’s [27] study has proposed that most flavonoids are precursors of anthocyanin synthesis. With the acceleration of flavonoid synthesis, anthocyanin content also increases. Flavonoids participate in the synthesis of multiple components and act as intermediates to buffer these components. In this study, the leaves of ‘Zhonghuahongye’ were analyzed using LC-MS/MS, and 273 metabolites were identified. These were primarily divided into eight categories, including anthocyanins, flavonoids, flavonol, flavanone, flavonoids, procyanidins, isoflavone and polyphenols. Among the three groups, G1 vs. G2, G1 vs. G3 and G2 vs. G3, the top 10 metabolites with the largest down-regulation differences were all flavonoids, anthocyanins, and flavonoids. The downregulation of anthocyanins primarily occurs during the transition from G1 to G2, which is similar to the period when the pigment content of anthocyanins sharply decreases. The differential metabolites are primarily enriched in four metabolic pathways: flavone and flavonol biosynthesis (ko00944), flavonoid biosynthesis (ko00941), isoflavonoid biosynthesis (ko00943) and anthocyanin biosynthesis (ko00942). The secondary differential metabolic pathway is significantly enriched in the anthocyanin biosynthesis pathway. Therefore, we speculate that the degradation of anthocyanins and the inhibition of anthocyanin synthesis primarily occurs in the transition period from G1 to G2, which is related to the transformation of flavonoids.

The biosynthesis of flavonoids is a complex process involving multiple structural genes and TFs. The biosynthesis of flavonoids is regulated by multiple structural genes, including phenylalanine ammonia (*PAL*), cinnamic acid-4-hydroxylase(*C4H*), flavanone-3-hydroxylase (F3H), etc. [28]. The lack of dihydroflavonol 4-reductase(*DFR*) activity is a limiting factor for anthocyanin accumulation [29], and UDP-glycose flavonoid glycosyltransferase (*UFGT*) is a key enzyme for anthocyanin synthesis. Improving *UFGT* enzyme activity can promote anthocyanin synthesis [30]. In our study, with the decrease in *DFR* and flavonol synthase(*FLS*) gene expression, we found that the content of peonidin *O*-hexoside and malvidin 3-*O*-galactoside decreased, leading to a decrease in anthocyanin content. With the decrease in anthocyanidin synthase (*ANS*) gene expression, we found that the content of cyanidin 3-*O*-glucoside decreased, leading to a decrease in anthocyanin content. Therefore, we speculate that the lack of activity of *DFR*, *ANS*, and *FLS* leads to a decrease in anthocyanins, thereby limiting the accumulation of anthocyanins. Flavonoid biosynthesis involves three types of TFs, namely *R2R3-MYB* factors, basic helix–loop–helix (*bHLH*) proteins, and protein WD40 repeats, forming complex interactions between *MYB-bHLH-WD40* TFs. In Yan et al.’s [31] study, *MYB* is found to be the primary determining factor for anthocyanin production, which is consistent with our results. Some studies have found that overexpressed *JMJ25* gene can bind to the transcription factor *MYB182* locus and end its chromatin expression, thus inhibiting the expression of structural gene in the anthocyanin biosynthesis pathway in poplar, causing a reduction in anthocyanin content, and negatively regulating the synthesis and accumulation of anthocyanins from the epigenetic level [32,33]. At present, research on ‘Zhonghuahongye’ is limited. In this experiment, differential metabolites (*DEM*), differential genes, and transcription factors related to anthocyanin synthesis were established in a network diagram. It was found that 434 transcription factors were differentially expressed simultaneously in the three combinations of G1 vs. G2, G1 vs. G3, and G2 vs. G3, including 13 *B3*, 32 *bHLH*, 18 *bZIP*, 31 *MYB*, 25 *NAC*, and 20 *WRKY* transcription factors. These 434 transcription factors are considered to be important factors in regulating the differences in anthocyanin synthesis during different developmental stages of Zhonghuahongye’s green leaves. It has been reported that *MYB* determines the amount of anthocyanin produced by specific cells and hence the variation in the color intensity and pattern of the flower [34]. The *MYB* transcription factors reported to be involved in the regulation of flavonoids and anthocyanins in *Arabidopsis thaliana* were *MYB111*, *MYB113*, *MYB114*, and *MYB123* genes. The *Populus trichocarpa MYB* transcription factors reported to be involved in the regulation of flavonoids and anthocyanins were *MYB5*, *MYB1*, *MYB111*, and *MYB119* genes. The *Vitis vinifera MYB* transcription factors reported to be involved in the regulation of flavonoids and anthocyanins were *MYB75*, *MYB12*, *MYB111*, *MYB113* and *R2R3* genes. At the same time, *MYB4*, *MYB5b*, *MYB113* and *R2R3* genes were *MYB* transcription factors, which are involved in the regulation of flavonoids and anthocyanins in *Zea mays*. Therefore, based on the differential expression of transcription factors and the important role of *MYB* transcription factors in flavonoid metabolism in different model plants, we chose *MYB* as the primary determining factor. We speculate that several key transcription factors may regulate the production of flavonoids and anthocyanins in ‘Zhonghuahongye’ [12,13,14]. Their function in flavonoid or anthocyanin biosynthesis still needs further research through functional analysis.

### 3.3. Atavism of Zhonghuahongye’s Leaves

Atavism refers to the occasional appearance of certain shapes of ancestors in organisms. Atavism is different from mutation and is a normal phenomenon in nature. The study of atavism in the plant kingdom is of great significance for understanding the evolution of plant organs. The atavism phenomenon is different in different plants and can occur in different parts of flowers, leaves, branches, etc. Additionally, the form of expression is also different, including flower type, flower color, or leaf color, leaf shape, or branch shape, gender, etc. At present, there are many studies on leaf color mutants in plant research [6], but the atavism of leaf color has not been systematically studied and needs to be further investigated. There are four principal conditions for the occurrence of atavism: accidental occurrence in nature, occurrence in sexual reproduction, induction of physical and chemical stimuli, and changes in environmental conditions [35]. According to previous studies on leaf atavism, the re-greening of leaves may substantially contribute to meeting carbon needs.

‘Zhonghuahongye’ is a poplar species with red leaves. During planting, the leaves turn green to varying degrees, which is called Zhonghuahongye’s leaf atavism. Cheng et al. [3] found that, among the retrogressive leaves of ‘Zhonghuahongye’, anthocyanin content in half-red, half-green and all-green leaves is lower than that of normal leaves, while the chlorophyll content is higher than that of normal leaves. The net photosynthetic rate is higher than that of normal leaves, which is consistent with the results of this study. In this study, we speculate that, to enhance the growth and development of plants and meet the needs of growth, practitioners should improve photosynthesis and inhibit the degradation of anthocyanins and the synthesis of anthocyanins. In this study, these actions led to a reduction in anthocyanin content in their leaves, the greening of the leaves, and atavism. In addition, the study of Hou [10] indicates that, in addition to having high photosynthetic activity in spring and autumn, the leaves also have a slow rate of water loss, and that good water retention also allows for the dilution of anthocyanins and a decrease in their content. Therefore, we speculate that changes in environmental conditions during the growth process, such as higher temperature and light, increased plant water retention, increased water content, and decreased anthocyanin content, may also cause the leaves to turn green and cause atavism. Subsequent research into atavism of leaves should conduct multi-angle measurement and analysis based on this reverse direction on such issues as the control of temperature and light intensity, the observation of morphological changes, changes in leaf tissue structure, the determination of pigment content, comparative analysis of proteome, and differences in gene expression. This study provides a theoretical basis for the follow-up study of leaf atavism.

## 4. Materials and Methods

### 4.1. Plant Material Treatments

In this study, three ‘Zhonghuahongye’ plants with strong growth were used. These were grown in Mengzhou Forest Farm in Xiguo Town, southwest of Mengzhou City, Henan Province (34°90′08″ N, 112°69′66″ E). This region has a warm temperate continental monsoon climate with four distinct seasons and an annual average temperature of about 13 °C, with the coldest weather being in January and the hottest weather in July. It always rains in summer, and the rainfall from June to August accounts for about 55% of the whole year. The annual average frost-free period lasts for about 210 days. The annual sunshine time totals approximately 2493 h. The dominant wind directions are northeast and southwest. In April 2019, we selected and sampled the leaves of ‘Zhonghuahongye’ on 1 April, 6 April and 11 April, respectively, and numbered the results as G1, G2 and G3, respectively. The sampling method was to select 1 bud-transformed branch at the same position of each plant (different individual plants were biological replicates), and 3 leaves were selected for each branch. The leaves were collected, placed in an ice box, and brought back to the laboratory to undergo immediate color phenotyping. Then, the leaves were quickly placed in liquid nitrogen.

### 4.2. Determination of Color Phenotype

The color phenotypes of the leaves in three periods were compared using an RAL Lucca colorimeter. The microstructure of the mesophyll, vein and petiole were analyzed via an optical microscope (OLYMPUS-BX51, Olympus corporation, Tokyo, Japan).

### 4.3. Determination of Pigment Content

The content of photosynthetic pigment in G1, G2 and G3 samples was determined via a direct extraction method. Researchers should clean and grind the blades (cut the veins) and weigh the blades with accuracy to 0.01 g. They should add 80% acetone 10 mL, soak samples in the dark for 72 h, add the supernatant, and extract chlorophyll. In our experiment, anthocyanins were extracted from the supernatant by grinding it with 10 mL 10% HCL methanol solution, leaving it for 4 h, and centrifuging it at 3500× *g* for 15 min. The light absorption values of the chlorophyll extract at 445 nm, 644 nm and 662 nm and anthocyanin extract at 530 nm and 657 nm were determined using an ultraviolet spectrophotometer (JINGHUA Instruments 752, Shanghai Jinghua Technology Instrument Co., Ltd, Shanghai, China). In total, 3 biological replicates and 3 technical replicates were detected for each sample. Anthocyanin content (C_A_), chlorophyll a content (C_a_), chlorophyll b content (C_b_), total chlorophyll content (C_T_) and carotenoid content (C_car_) were calculated according to Formulas (1)–(5):Chlorophyll a content (mg/g):C_a_ = (9.78 × A_662_ − 0.99 × A_664_)V/(W × 1000)(1)
Chlorophyll b content (mg/g):C_b_ = (21.40 × A_644_ − 4.65 × A_662_)V/(W × 1000)(2)
Total Chlorophyll content (mg/g):C_T_ = C_a_ + C_b_(3)
Carotenoid content (mg/g):C_car_ = (4.69 × A_445_)V/(W × 1000) − 0.27 × (C_a_ + C_b_)(4)
Anthocyanin content (mg/g):C_A_ = (A_530_ − 0.25×A_657_)V/(W × 1000)(5)

Note: A—light absorption value; V—extraction liquid volume; W— Sample quality.

### 4.4. Metabolite Analysis

The metabolites were extracted, detected and identified by the Met Ware Biotechnology Co., Ltd. (Wuhan, China). We processed mass spectrum data using Analyst 1.6.3. Qualitative analysis was carried out using the MWDB (Metware database) self-built database and public database of metabolite information, and the metabolite structure was analyzed by referring to MassBank, KNAPSAcK, HMDB, MoTo DB, METLIN and other existing public mass spectrometry databases. We quantified metabolites after obtaining the analytical data of metabolic mass spectra in different samples using the multi-reaction monitoring mode (MRM) from triple four-bar mass spectrometry. Then, the peak area integral of all mass spectra of substances was determined to ensure the accuracy of qualitative and quantitative measurements. The sample extracts were mixed and prepared into quality-control samples (QC) for use in quality-control analysis. The total ion flow pattern obtained by the essential spectra of different QC samples can be overlapped to judge the repeatability (technical repetition) of the experimental operation.

Metabolome data analysis included principal component analysis (PCA), cluster analysis (HCA), fold-change analysis, orthogonal partial least square discriminant analysis (OPLS-DA), differential metabolite screening, differential metabolite KEGG function annotation, and enrichment analysis [36].

Differential metabolites were screened on the basis of VIP value combined with differential multiple values of the OPLS-DA model, according to the following criteria:(1)If the metabolite difference is more than 2 times or less than 0.5 times, the difference is considered to be significant.(2)On the basis of the above, select metabolites with VIP ≥ 1. This indicates that the difference between groups of metabolites has a strong influence on the classification and discrimination of each group of samples, and that the difference is considered significant.

### 4.5. Transcriptomic Analysis

The transcriptomic analysis includes transcriptome sequencing and bioinformatics analysis. The RNA detected in this experiment refers to mRNA. Differential expression genes (DEGs) were analyzed, and functional annotation and enrichment analysis of DEGs were performed according to the database. The experimental process of transcriptome sequencing includes RNA extraction and detection, library construction and computer sequencing.

Transcriptome sequencing was performed, whereby we used the FastPure^®^ Plant Total RNA Isolation Kit (Vazyme RC401, Vazyme Biotech Co., Ltd., Nanjing, China) to extract total RNA from leaves. mRNA was obtained via the enrichment or removal of ribosomal RNA from total RNA by oligo (dT) magnetic beads. The first cDNA and second cDNA were synthesized using the fragment RNA interrupted by the fragmentation buffer as the template. The quality of the constructed library was tested, and then the qualified cDNA library was sequenced by the Illumina HiSeq platform (Wuhan Metwell Biotechnology Co., Ltd., Wuhan, China).

Bioinformatics analysis used HISAT2 [37] to sequence the filtered high-quality clean data with the reference genome of Populus trichocarpa [38]. FPKM was used as the data to describe gene expression levels, and inter-sample (biological duplication) correlation and PCA were performed according to expression levels. The differentially expressed gene sets were screened according to |log2Fold Change| ≥ 1 and FDR < 0.05 [39,40].

Functional annotation and enrichment analysis of differentially expressed genes were performed using KEGG [41], GO, and KOG [42,43] databases. The iTAK 1.6 software was used to predict plant transcription factors, and hmmscan was compared with the defined transcription factors in PlnTFDB and planted databases to identify transcription factors.

We performed qRT-PCR analysis on genes related to anthocyanin synthesis. In this experiment, the FastPure^®^ Plant Total RNA Isolation Kit (Vazyme RC401, Vazyme Biotech Co., Ltd., Nanjing, China) was used to extract total RNA from leaves according to the kit instructions. The integrity of the bands was detected via 1.0% agar-gel electrophoresis with an ultramicrospectrophotometer to determine the total RNA quality. The first strand of cDNA was synthesized according to the instructions of the HiScript^®^ III All-in-one RT SuperMix Perfect for the qPCR reverse transcription kit (Vazyme R333, Vazyme Biotech Co., Ltd., Nanjing, China). A 20 µL reaction system was used for reverse transcription. A 20 µL reaction system was used for real-time fluorescence quantitative analysis. The specific primers of 10 screening genes were designed using Oligo7 in the qRT-PCR assay. All primers are shown in Appendix A. Taq Pro Universal SYBR qPCR Master Mix Kit (Vazyme Q712-02, Vazyme Biotech Co., Ltd., Nanjing, China) was used for qRT-PCR analysis according to the kit instructions, and EF418792 was used as the internal reference gene. The relative expression of genes was calculated using 2^−ΔΔCT^ [44,45].

## 5. Conclusions

This study focused on the diversity of flavonoids in the green leaves of ‘Zhonghuahongye’. The phenotypic results showed that the color of the leaves changed from copper-brown to green-brown and finally to fern green during the growth process. ‘Zhonghuahongye’ naturally contains anthocyanins. The anthocyanin content of leaves decreases significantly (by 62.5%) with growth, while the chlorophyll content increases. The chlorophyll a content increases by 204.35%, the chlorophyll b content increases by 69.23%, the total chlorophyll content increases by 155.56%, and the carotenoid content increases by 60%. Combined with metabolite and gene expression profiling, the mechanism of color fading of red leaves was clarified. Furthermore, 273 different modified flavonoids were detected using widely targeted metabolomics. These flavonoids decrease with the maturity of leaves. Anthocyanins are the primary coloring substances that turn red during the green leaf tender period. The obvious decrease in the content of Cyanidin 3-*O*-glucoside, peonidin *O*-hexoside and malvidin 3-*O*-galactoside in the leaves with growth and development may be the reason for the disappearance of red in the delicate leaves. The KEGG functional enrichment analysis shows that the differential metabolites and differential genes are primarily enriched via the biosynthesis of secondary metabolites (ko01110), metabolic pathway (ko01100), isoflavonoid biosynthesis (ko00943), anthocyanin biosynthesis (ko00942), flavonoid biosynthesis (ko00941), and flavone and flavonol biosynthesis (ko00944). *DFR*, *FLS* and *ANS* downstream of anthocyanin synthesis may be the key structural genes for lowering anthocyanin synthesis and accumulation in Zhonghuahongye’s green leaves. Meanwhile, 390 important transcription factors were screened for the differences in anthocyanin synthesis during the different developmental stages of green leaves in ‘Zhonghuahongye’. Multiomics analysis shows that the formation of color is primarily influenced by the regulation of anthocyanins and their related synthesis influencing genes.

To summarize, we studied the transcriptome and metabolic response of Zhonghuahongye’s leaves during the color fading of red leaf regression. During this process, many genes and metabolites underwent changes. These results provide a theoretical basis for the screening of important genes and metabolites in ‘Zhonghuahongye’.

## Figures and Tables

**Figure 1 plants-12-03511-f001:**
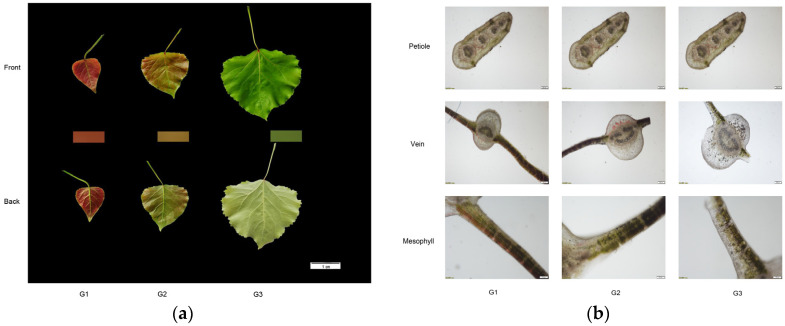
Comparison of leaf color and phenotype changes: (**a**) leaf phenotypic color determination at different periods (scale bar: 1cm); (**b**) microstructure of leaves at different periods(scale bar: Petiole: 200 μm; Vein: 200 μm; Mescophyll: 100 μm). Note: G1: 1 April; G2: 6 April; G3: 11 April.

**Figure 2 plants-12-03511-f002:**
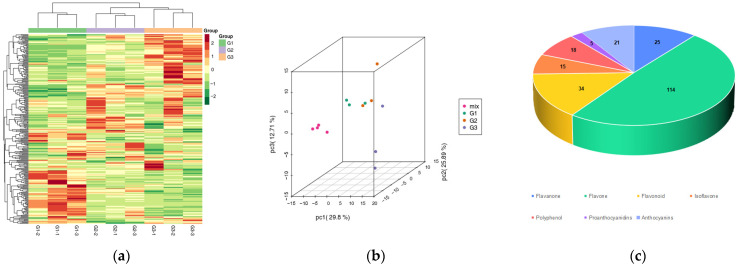
Qualitative and quantitative analysis of leaf metabolites in different periods: (**a**) sample aggregate metabolite cluster diagram; (**b**) PCA score chart of quality spectrum data of samples and quality control samples for each group; (**c**) sample aggregate metabolite clustering diagram.

**Figure 3 plants-12-03511-f003:**
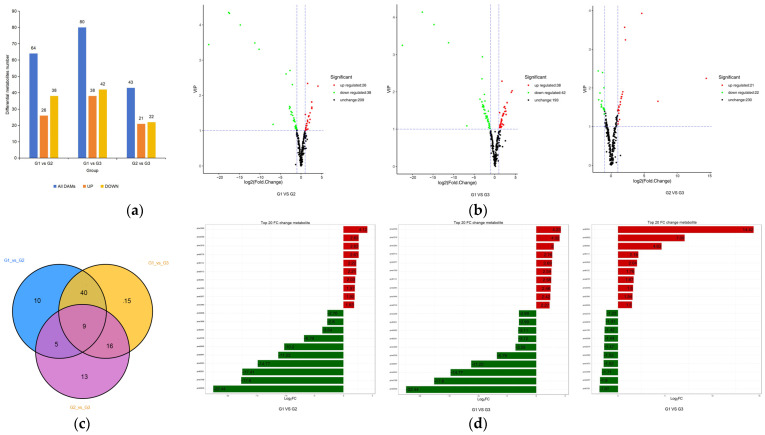
Analysis of leaf metabolites difference in different periods: (**a**) the amount of leaf metabolites in different periods; (**b**) differential metabolite volcano map; (**c**) Venn diagram; (**d**) histogram of metabolite difference multiples.

**Figure 4 plants-12-03511-f004:**
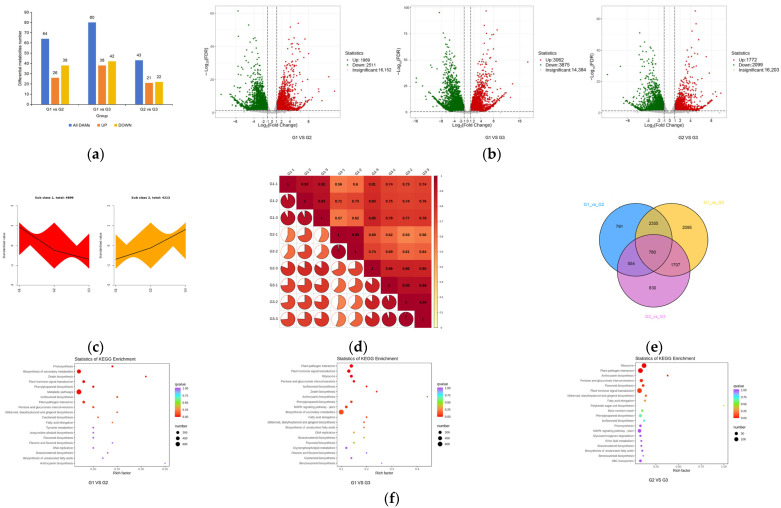
Leaf transcriptome analysis of ‘Zhonghuahongye’ at different periods: (**a**) differential gene statistical map; (**b**) differential gene volcano map; (**c**) k-means cluster graph; (**d**) correlation heat map; (**e**) Venn diagram; (**f**) differential gene KEGG enrichment map.

**Figure 5 plants-12-03511-f005:**
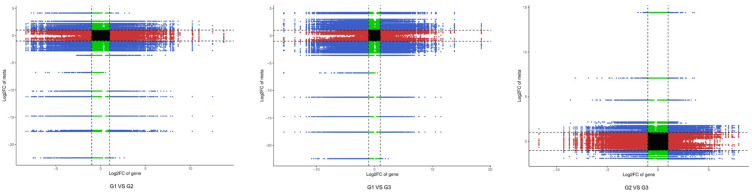
Nine-quadrant diagram of correlation analysis. Note: The horizontal coordinate indicates the log2FC of the gene and the vertical coordinate indicates the log2FC of the metabolite.

**Figure 6 plants-12-03511-f006:**
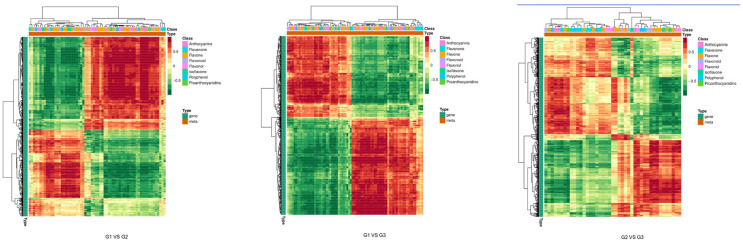
Correlation clustering heatmap. Note: In the figure, each row has a gene, each column is a metabolite, red represents a positive correlation between the gene and metabolite, and green represents a negative correlation between the gene and the metabolite.

**Figure 7 plants-12-03511-f007:**
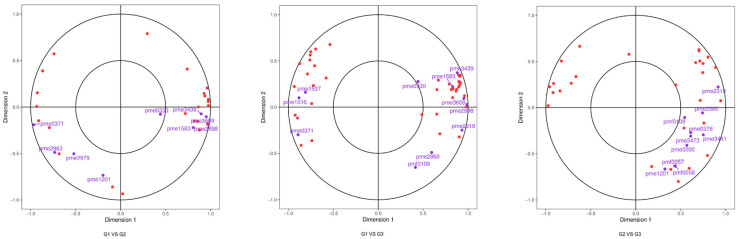
CCA diagram. Note: In the figure, four regions are distinguished by crosses. In the same region, the farther away from the origin a point was, the closer the distance between each other and the higher the correlation. Metabolites are labeled purple and genes are labeled red. If there is too much of a certain type of substance, in order to avoid overlapping text, it will be displayed with dots instead.

**Figure 8 plants-12-03511-f008:**
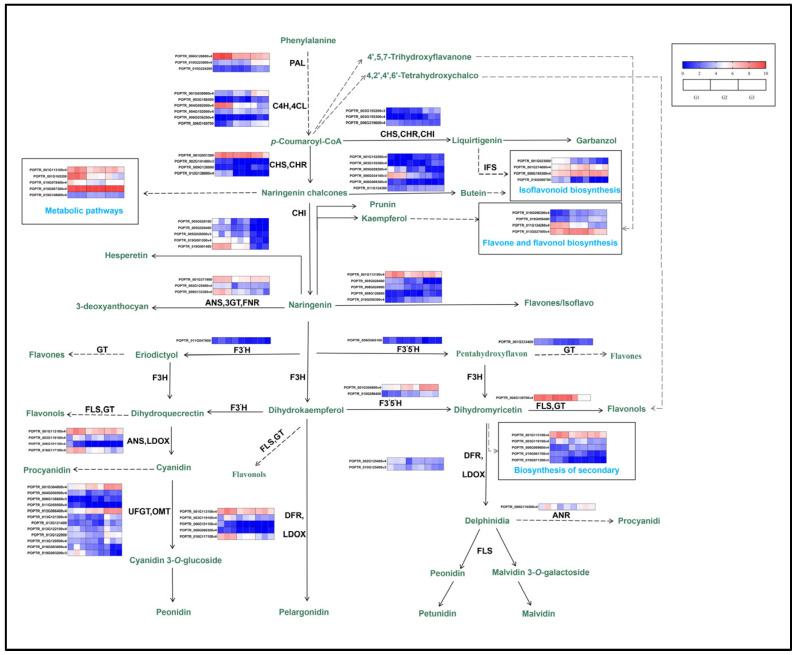
Bioanabolism of anthocyanins in the leaves of ‘Zhonghuahongye’. Note: Each color circle represents the level of gene expression according to the color scale. The enzymes bioanabolically associated with anthocyanins of ‘Zhonghuahongye’ are shown in Appendix A. Detailed information for 83 differentially expressed genes is shown in Appendix A.

**Table 1 plants-12-03511-t001:** Analysis of pigment content of leaves during different periods (mg/g).

Pigment	G1	G2	G3
Anthocyanin	0.24 ± 0.00 a	0.10 ± 0.01 b	0.09 ± 0.00 c
Chlorophyll a	0.23 ± 0.00 b	0.71 ± 0.03 a	0.70 ± 0.03 a
Chlorophyll b	0.13 ± 0.01 b	0.22 ± 0.01 a	0.22 ± 0.01 a
Total chlorophyll	0.36 ± 0.01 b	0.93 ± 0.04 a	0.92 ± 0.04 a
Carotenoid	0.15 ± 0.02 b	0.25 ± 0.02 a	0.24 ± 0.02 a

Note: Different lowercase letters: significant differences (*p* < 0.05)

**Table 2 plants-12-03511-t002:** Functional annotation analysis of differential metabolites KEGG in different metabolic pathways (pieces).

KEGG Pathway	G1 vs. G2	G1 vs. G3	G2 vs. G3
ko00944	2	1	1
ko00943	7	7	6
ko00942	2	3	2
ko00941	9	11	10

**Table 3 plants-12-03511-t003:** Analysis of KEGG enrichment results.

KEGG_Map	Description	*p*-Value_Meta	Count_Meta	*p*-Value_Gene	Count_Gene
ko01110	Biosynthesis of secondary metabolites	0.14436567	11	4.200984 × 10^−6^	470
ko01100	Metabolic pathways	0.76832403	3	5.180148 × 10^−5^	793
ko00943	Isoflavonoid biosynthesis	0.04405258	7	2.393100 × 10^−4^	19
ko00942	Anthocyanin biosynthesis	0.45535157	2	1.330260 × 10^−2^	4
ko00941	Flavonoid biosynthesis	0.25504294	9	1.581373 × 10^−2^	24
ko00944	Flavone and flavonol biosynthesis	0.93960917	2	2.652742 × 10^−2^	7

**Table 4 plants-12-03511-t004:** Common regulatory genes in different development stages of green leaves of ‘Zhonghuahongye’.

ID	Family	Category	log2 FC	Type
POPTR_017G082500v4	MYB	TF	2.952	up
POPTR_017G125600	MYB	TF	−3.548	down
POPTR_015G082700v4	MYB	TF	6.074	up
POPTR_014G081200v4	MYB	TF	3.203	up
POPTR_013G149200v4	MYB	TF	−3.054	down
POPTR_012G080400v4	MYB	TF	−3.344	down
POPTR_010G141000v4	MYB	TF	−5.133	down
POPTR_009G053900v4	MYB	TF	4.733	up
POPTR_003G114100	MYB	TF	3.972	up
POPTR_001G118800v4	MYB	TF	3.289	up

## Data Availability

All authors agree with MDPI Research Data Policies.

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
