# Peer review of "Comparative Transcriptome and Metabolome Profiling Reveal Mechanisms of Red Leaf Color Fading in *Populus* × *euramericana* cv. ‘Zhonghuahongye’"

_plants, 2023, doi:10.3390/plants12193511_

Round 1
Reviewer 1 Report
本文通过转录组和代谢组学分析揭示了中国红叶杨叶的褪色机制,筛选了可能在红叶凋谢过程中起关键作用的差异表达基因。对后续中国红叶杨叶褪色机理的研究具有一定意义。
文章的研究方向有优点,但也有个别问题。
1. 比较转录组和代谢组谱揭示了胡杨×欧美cv.'中华红叶'红叶褪色的机理 本文中,但创新点不明确,在前期研究的基础上如何做出何种改进,建议深入寻找创新点,提高研究的意义部分。
2. 本稿仅对三个不同生长阶段的叶片进行了转录组和代谢组学分析,未涉及任何实验验证。实验证据不足,实验结果的可靠性值得怀疑。
3. 许多参考文献都有类似的内容。建议整理文章中引用参考文献的部分,删除一些太旧、没有参考价值的部分。
4. 图1中不同阶段叶片G2、G3、G1的分类标准是什么?请提供详细说明。
5.图1中,G2期间图B与其他两个的拍摄角度存在显著差异。请问是不是由于G2时期特殊的叶形态?如果没有,请说明原因并补充照片,以保持不同时期的相同拍摄角度。
6、图注说明不够具体,建议在每个图注中突出结果部分,使图的内容清晰。
7.每个图像的字体大小和格式各不相同,因此我们建议将其修改为适当的大小。
8、稿件中出现的基因应为斜体,建议对整篇稿件进行梳理,对本段内容进行修改。
接受重大修订
(from Google Translate)
This article reveals the fading mechanism of Chinese red poplar leaves through transcriptome and metabolomic analysis, and screens differentially expressed genes that may play a key role in the withering process of red leaves. It is of certain significance for the subsequent research on the fading mechanism of Chinese red poplar leaves.
The research direction of the article has advantages, but there are also individual problems.
1. Comparing transcriptome and metabolome profiles reveals the mechanism of red leaf fading of Populus euphratica point to improve the significance of the research.
2. This manuscript only conducted transcriptome and metabolomics analysis on leaves at three different growth stages, and did not involve any experimental verification. The experimental evidence is insufficient and the reliability of the experimental results is questionable.
3. Many references have similar content. It is recommended to organize the referenced parts of the article and delete some parts that are too old and have no reference value.
4. What are the classification criteria for leaves G2, G3, and G1 at different stages in Figure 1? Please provide detailed instructions.
5. In Figure 1, there is a significant difference between the shooting angles of Figure B during G2 and the other two. Is it due to the special leaf shape in the G2 period? If not, please explain why and add photos to maintain the same shooting angle from different time periods.
6. The legend description is not specific enough. It is recommended to highlight the result part in each figure legend to make the content of the figure clear.
7. Each image has a different font size and format, so we recommend modifying it to an appropriate size.
8. The genes appearing in the manuscript should be in italics. It is recommended to sort out the entire manuscript and modify the content of this paragraph.
Major revisions
本文语言流畅,逻辑清晰,没有明显的语言错误。
(from Google Translate)
The language of this article is fluent, the logic is clear, and there are no obvious language errors.
Reviewer 2 Report
The manuscript by Shaowei et al. uses biochemical, metabolic, and bioinformatic approaches to investigate the mechanism of color fading in red leaf of Chinese red Poplar. The authors have generated a solid amount of data and the conclusions are justified by the results. However, some experiments and several details need to be clarified in the text or legends.
1. In Results part.
Line 85: It is better to define what are G1/2/3 periods in the first appearance.
Figure 1: include scale bars in legends.
Figure 2: Add a title for the whole figure.
Line 132: Full name of “OPLS-DA”.
Figure 3: Add a title for the whole figure.
Line 237: “KMGG”, KEGG?
Line 357: “It mainly has 6 kinds of non-ligands (Aglycone) including Pelargonidin, Cyanidin, Delphinidin, Petunidin and Malvidin”, miss Peonidin.
Figure 8: Include details in legends, such as: full names of enzymes, the meaning of color intensity, etc.
2. In Discussion part.
Line 439: “the decrease in anthocyanin content of 'Zhonghuahongye'. at different stages is due to the inhibition of anthocyanin degradation and synthesis”, what that means? Inhibition of synthesis and enhance degradation?
Line 690 and Figure S4: “KMGG”, KEGG?
Figure S7: what is BZ1? And no primer sequences in Table S1.
Table S1: Do not match with Figure S7. Where is the results of WER, MTB74(MYB74).
Reviewer 3 Report
Review Plants 2606248
Chinese red poplar Populus × euramericana cv. 'Zhonghuahongye' with dark red leaf color provides excellent landscape effect. However, color fading of red leaves during cultivation presents a problem for its application in leaf ornamental woods. The authors have used transcriptomic and metabolomic analyses to unravel the molecular mechanisms underlying leaf greening. According to the results of the study, downregulation of key anthocyanin synthesis genes DFR, FLS and ANS may be responsible for metabolic changes causing alterations in types and proportions of pigments. Among them the decrease in the contents of cyanidin 3-O-glucoside, peonidin O-hexoside and malvidin 3-O-galactoside in the growing leaves could be the cause of the disappearance of 'Zhonghuahongye'’red leaves
Overall, the authors presented valuable data confirming the role of anthocyanins and their related synthesis genes in color changes of leaves in Chinese red poplar. However, possible function of transcription factors limiting the accumulation of anthocyanins deserves more thorough discussion, as well as the nature of the so called “leaf atavism” which according to the authors is the basis of color changes related to the environment and ontogenetic factorsю
In addition, despite the undoubted advantages and a large volume of presented results which are meticulously described in the manuscript, the submitted version cannot be accepted for publication, as extensive editing of the English language and style is required. The total number of illiterate and incomprehensible sentences and the inappropriate use of capital letters exceeds the permissible for a detailed enumeration, I will give just one example:
Lines500-502:
“Flavonoid biosynthesis is composed of three types of TFs, namely R2R3-MYB factors, basic helix loop elix (bHLH) proteins, and protein WD40 repeats, forming complex interactions between MYB-bHLH-WD40.” It would be just as good if the authors also argued why in their study, “MYB is the main determining factor for anthocyanin production” (line 503).
In addition, the overall impression of the manuscript would be improved if the authors avoided numerous repetitions when describing the results or omitted information that the RNA detected in the transcriptomic analysis refers to mRNA
Extensive editing of the English language and style is required.
Author Response
请参阅附件。

Round 2
Reviewer 3 Report
Review 2 Plants 2606248
The additions made by the authors contributed to improving the overall impression of the manuscript They more accurately reflect the authors' ideas about the mechanisms of color fading in Populus. I think the manuscript can be accepted for publication. However, some minor corrections still need to be made, in particular.
Line 175” in different stages” to “at different stages”
Line 276: “were” to “was”
Line 346: “determines” to “determine”
Line 364: “tissue” to “organ”
Line 375 “are” should be inserted before “direct cause”
Line 407: “in the early stage” to “at the early stage”
Line 410: “leaf antibodies”, perhaps, to “leaf sizes”?
Line 433 «Transcriptome” to “transcriptome”
Line 461 «Flavonoids” to “flavonoids”
Line 481: “gene” to “genes”
Minor editing of English language is still required
